# Crystal structure of *Drosophila* Piwi

Sonomi Yamaguchi[1,6], Akira Oe[1,6], Kazumichi M. Nishida[2,6], Keitaro Yamashita [1,6], Asako Kajiya[2], Seiichi Hirano[1], Naoki Matsumoto[1], Naoshi Dohmae[3], Ryuichiro Ishitani[1], Kuniaki Saito[4], Haruhiko Siomi[5], Hiroshi Nishimasu[1✉], Mikiko C. Siomi[2✉] & Osamu Nureki [1✉]

PIWI-clade Argonaute proteins associate with PIWI-interacting RNAs (piRNAs), and silence transposons in animal gonads. Here, we report the crystal structure of the *Drosophila* PIWI-clade Argonaute Piwi in complex with endogenous piRNAs, at 2.9 Å resolution. A structural comparison of Piwi with other Argonautes highlights the PIWI-specific structural features, such as the overall domain arrangement and metal-dependent piRNA recognition. Our structural and biochemical data reveal that, unlike other Argonautes including silkworm Siwi, Piwi has a non-canonical DVDK tetrad and lacks the RNA-guided RNA cleaving slicer activity. Furthermore, we find that the Piwi mutant with the canonical DEDH catalytic tetrad exhibits the slicer activity and readily dissociates from less complementary RNA targets after the slicer-mediated cleavage, suggesting that the slicer activity could compromise the Piwi-mediated co-transcriptional silencing. We thus propose that Piwi lost the slicer activity during evolution to serve as an RNA-guided RNA-binding platform, thereby ensuring faithful co-transcriptional silencing of transposons.

[1] Department of Biological Sciences, Graduate School of Science, The University of Tokyo, 7-3-1 Hongo, Bunkyo-ku, Tokyo 113-0033, Japan. [2] Department of Biological Sciences, Graduate School of Science, The University of Tokyo, 2-11-16 Yayoi, Bunkyo-ku, Tokyo 113-0032, Japan. [3] Biomolecular Characterization Unit, RIKEN Center for Sustainable Resource Science, 2-1 Hirosawa, Wako, Saitama 351-0198, Japan. [4] Invertebrate Genetics Laboratory, National Institute of Genetics, 1111 Yata, Mishima, Shizuoka 411-8540, Japan. [5] Keio University School of Medicine, 35 Shinanomachi, Shinjuku-ku, Tokyo 160-8582, Japan.. [6] These authors contributed equally: Sonomi Yamaguchi, Akira Oe, Kazumichi M. Nishida, Keitaro Yamashita ✉email: nisimasu@bs.s.u-tokyo.ac.jp; siomim@bs.s.u-tokyo.ac.jp; nureki@bs.s.u-tokyo.ac.jp

Argonaute proteins bind small non-coding RNA guides to form RNA-induced silencing complexes, which recognize target RNAs complementary to the guide RNAs[1, 2]. The Argonaute proteins can be divided into two clades, AGO and PIWI. The AGO-clade Argonautes (AGOs) are ubiquitously expressed and bind 20–22-nucleotide (nt) microRNAs or small interfering RNAs, to regulate gene expression. The PIWI-clade Argonautes (PIWIs) bind 24–31-nt PIWI-interacting RNAs (piRNAs) to form piRNA-induced silencing complexes, which silence transposons and maintain genome integrity in animal gonads[3–6].

*Drosophila melanogaster* has three PIWIs: Piwi, Aubergine (Aub), and Argonaute3 (Ago3). Aub and Ago3 are RNA-guided RNases (slicers) that cleave transposon transcripts and piRNA precursor transcripts in the cytoplasm, thereby coupling the piRNA production and transposon silencing[7,8]. In contrast, Piwi co-transcriptionally silences transposons in the nucleus, by facilitating heterochromatin formation at target transposon loci[9–12]. The Piwi–piRNA complex interacts with nascent transposon transcripts and essential cofactors, such as the zinc-finger protein Gtsf1/Asterix (Arx)[13–15] and the adaptor protein Silencio/Panoramix (Panx)[16–19]. The H3K9 methyltransferase Eggless/SetDB1 and the H3K4 demethylase Lsd1/Su(var)3-3 are then targeted to the transposon loci, thereby establishing heterochromatin[18,19]. Recent studies have shown that the RNA-binding protein Nxf2 interacts with Panx and Nxt1/p15, and reinforces the association of the Piwi–piRNA complex with the target transcripts[20–23]. In addition, the Piwi-mediated silencing requires other cofactors, such as the heterochromatin-binding protein HP1a[9], the RNA-binding protein Maelstrom[10], and the linker histone H1[24].

Previous structural studies provided mechanistic insights into the Argonaute-mediated RNA silencing[25–28]. The crystal structures of the eukaryotic AGOs, such as *Kluyveromyces polysporus* Ago (KpAgo)[29], human Ago1[30,31], Ago2 (hAgo2)[32–34], Ago3[35], and Ago4[36], revealed that Argonautes adopt a bilobed architecture consisting of four signature domains (N, PAZ, MID, and PIWI) and two linker domains (L1 and L2), in which the 3′ and 5′ ends of the guide RNA are recognized by the PAZ and MID-PIWI domains, respectively. The crystal structures also demonstrated that the PIWI domain adopts an RNaseH fold, with the DEDX (X is usually H or D) catalytic tetrad responsible for the target cleavage. Furthermore, the crystal structures of prokaryotic AGOs, such as *Pyrococcus furiosus* Ago (PfAgo)[37], *Thermus thermophilus* Ago (TtAgo)[38–41], *Rhodobacter sphaeroides* Ago (RsAgo)[42,43], *Marinitoga piezophila* Ago (MpAgo)[44,45], *Methanocaldococcus jannaschii* Ago (MjAgo)[46], and *Clostridium butyricum* Ago (CbAgo)[47], highlighted the mechanistic conservation and divergence between the eukaryotic and prokaryotic AGOs. In particular, the eukaryotic and prokaryotic AGOs generally use a conserved lysine residue and a metal ion to recognize the 5′-phosphate group of their guide strand, respectively.

In contrast to AGOs, the structural information on PIWIs has been limited, primary owing to difficulties in protein preparation. Nonetheless, we recently purified the silkworm PIWI protein Siwi bound to endogenous piRNAs from silkworm ovary-derived, cultured BmN4 cells, using a monoclonal anti-Siwi antibody, and determined the crystal structure of the Siwi–piRNA complex[48]. A structural comparison of Siwi with the AGO-clade Argonautes highlighted notable variations in the spatial arrangement of the N-PAZ lobes with respect to the MID-PIWI lobes, reflecting their functional differences. The structure further revealed that Siwi recognizes the piRNA 5′ phosphate in a metal-dependent manner, as in the prokaryotic AGOs. However, the extent of the conservation of these structural features among the PIWI-clade Argonautes remains enigmatic, as the structural information

about PIWIs has been limited to Siwi. In addition, no study has explicitly assessed whether Piwi exhibits or lacks slicer activity in vitro.

In this study, we purified the endogenous Piwi–piRNA complex from cultured fly ovarian somatic cells (OSCs)[49], and determined the crystal structure of the Piwi–piRNA complex at 2.9 Å resolution. A structural comparison between Piwi and Siwi indicated that the structural features, such as the N domain orientation and the metal-dependent piRNA recognition, are conserved among the PIWI-clade Argonautes. Our structural and biochemical data demonstrated that Piwi has a DVDK, rather than DEDH, tetrad in the PIWI domain, and lacks slicer activity in vitro. Furthermore, we found that the Piwi mutant with the canonical DEDH catalytic tetrad displays the slicer activity and dissociates from partially complementary RNA targets after the slicer-mediated cleavage, suggesting that the slicer activity could compromise the Piwi-mediated co-transcriptional silencing. Overall, our findings provide a critical step toward a mechanistic understanding of Piwi-mediated transposon silencing.

## Results

**Piwi preparation**. The anti-Piwi monoclonal antibody P3G11 recognizes the N-terminal disordered region of Piwi[50]. We immunoisolated the endogenous Piwi from OSCs[49], using Sepharose beads conjugated with the anti-Piwi antibody, and then treated the mixture with chymotrypsin, as described previously[51] (Supplementary Fig. 1a). An sodium dodecyl sulphate-polyacrylamide gel electrophoresis analysis of the supernatant showed that, whereas the immunoisolated full-length Piwi (97 kDa) migrated as an ~ 100-kDa band on the gel, a slightly smaller band was liberated in the supernatant fraction after the chymotrypsin treatment (Supplementary Fig. 1b). The immunoisolated Piwi was further purified by chromatography on heparin and size-exclusion columns (Supplementary Fig. 1c and 1d). An N-terminal sequence analysis revealed that the purified Piwi begins with Arg34. The ratio of the absorbances at 260 and 280 nm of the peak fraction indicated that the purified Piwi associates with nucleic acids (Supplementary Figure 1c). Indeed, $^{32}$P-end labeling of the RNAs revealed that about 26-nt RNAs are associated with Piwi (Supplementary Fig. 1e), consistent with previous studies[7,8,49,50]. These results indicated that the purified sample represents the Piwi–piRNA complex core (residues 34–843) lacking the flexible N-terminal region.

**Overall structure**. We determined the crystal structure of the Piwi–piRNA complex at 2.9 Å resolution by molecular replacement, using the Siwi structure (PDB: 5GUH)[48] as the search model (Table 1). The structure revealed that Piwi consists of four domains (N, PAZ, MID, and PIWI) and three linker regions (L0, L1, and L2), and it can be divided into two lobes (N-PAZ and MID-PIWI) (Fig. 1a and 1b). The N-PAZ lobe consists of the L0 (residues 103–113), N (residues 114–186), L1 (residues 187–263), PAZ (residues 264–371), and L2 (residues 372–429) domains, whereas the MID-PIWI lobe consists of the L0 (residues 93–102), L2 (residues 430–470), MID (residues 471–601), and PIWI (residues 602–843) domains.

We observed relatively poor electron densities for the PAZ domain (Supplementary Fig. 2a and 2b), indicating that the PAZ domain is flexible in the present structure. The PAZ domain of Piwi shares 40% sequence identity with that of Siwi, indicating that the PAZ domains of Piwi and Siwi are structurally similar. Thus, we fitted a Siwi-based homology model into the electron density, and then refined the model using the secondary-structure restraints. The resulting model is consistent with the anomalous difference peaks for two putative mercury ions (derived from the

## Table 1 Data collection and refinement statistics.

| Data collection | |
|---|---|
| Space group | $P2_12_12_1$ |
| Cell dimensions | |
| $a, b, c$ (Å) | 62.1, 115.6, 119.9 |
| $\alpha, \beta, \gamma$ (°) | 90, 90, 90 |
| Resolution (Å)* | 39.93–2.90 (3.08–2.90) |
| $R_{meas}$*,** | 0.551 (21.2) |
| $<I/\sigma(I)>$*,** | 10.1 (0.52) |
| $CC_{1/2}$*,** | 0.998 (0.676) |
| Completeness (%)*,** | 100 (100) |
| Redundancy*,** | 74.4 (75.5) |
| **Refinement** | |
| Resolution (Å) | 39.90–2.90 |
| No. reflections | 19,774 |
| $R_{work}/R_{free}$ | 0.2394 / 0.2585 |
| No. atoms | |
| Protein | 5,402 |
| RNA | 87 |
| Ion | 15 |
| Averaged B-factors (Å$^2$) | |
| Protein | 110.4 |
| RNA | 97.5 |
| Ion | 132.3 |
| R.m.s. deviations from ideal | |
| Bond lengths (Å) | 0.002 |
| Bond angles (°) | 0.566 |
| Ramachandran plot | |
| Favored (%) | 95.01 |
| Allowed (%) | 4.28 |
| Outlier (%) | 0.71 |

*Values in parentheses are for the highest-resolution shell.
**Friedel pairs are treated as different reflections.

crystallization solution), which are located in the vicinities of Cys271 and Cys317 (Supplementary Fig. 2c).

We observed electron densities that could be fitted to the co-purified endogenous piRNAs (Supplementary Fig. 2d), as in the previous Argonaute structures, such as hAgo2[33] and Siwi[48]. The electron density for the 5′ nucleotide could be fitted to a uridine, consistent with the preference of Piwi for the U1 nucleotide in the bound piRNA[7,8,49,50]. Electron densities were not observed for the other nucleotides of the piRNAs, suggesting that they are disordered in the present structure. Thus, we modeled UAUU as the 5′ nucleotides of the bound piRNAs, according to the size and shape of the densities.

**Structural comparison**. A structural comparison of Piwi with Siwi[48] (Fig. 2a) and hAgo2[34] (Fig. 2b) highlighted the similarities and differences between PIWIs and AGOs. Their individual domains superimposed well (root-mean-square deviation of 0.9–2.0 Å for equivalent Cα atoms), although they share limited sequence identities (Supplementary Fig. 3). The superimposition of Piwi on Siwi and hAgo2, based on their PIWI domains, revealed that the arrangements of the N domains relative to the L1 domain are similar between Piwi and Siwi, but different between Piwi/Siwi and hAgo2 (Piwi adopts a slightly open conformation as compared to Siwi, due to the different orientation of their α-helices at the interface) (Figs. 2c, d). The distinct arrangements are stabilized by hydrophobic interactions at the interfaces between the N and L0-L1-L2 domains. Notably, two tryptophan residues in the L0 and L2 domains (Trp112/Trp422 in Piwi and Trp166/Trp477 in Siwi) are highly conserved among the PIWIs, but not among the AGOs (Supplementary Fig. 4), and

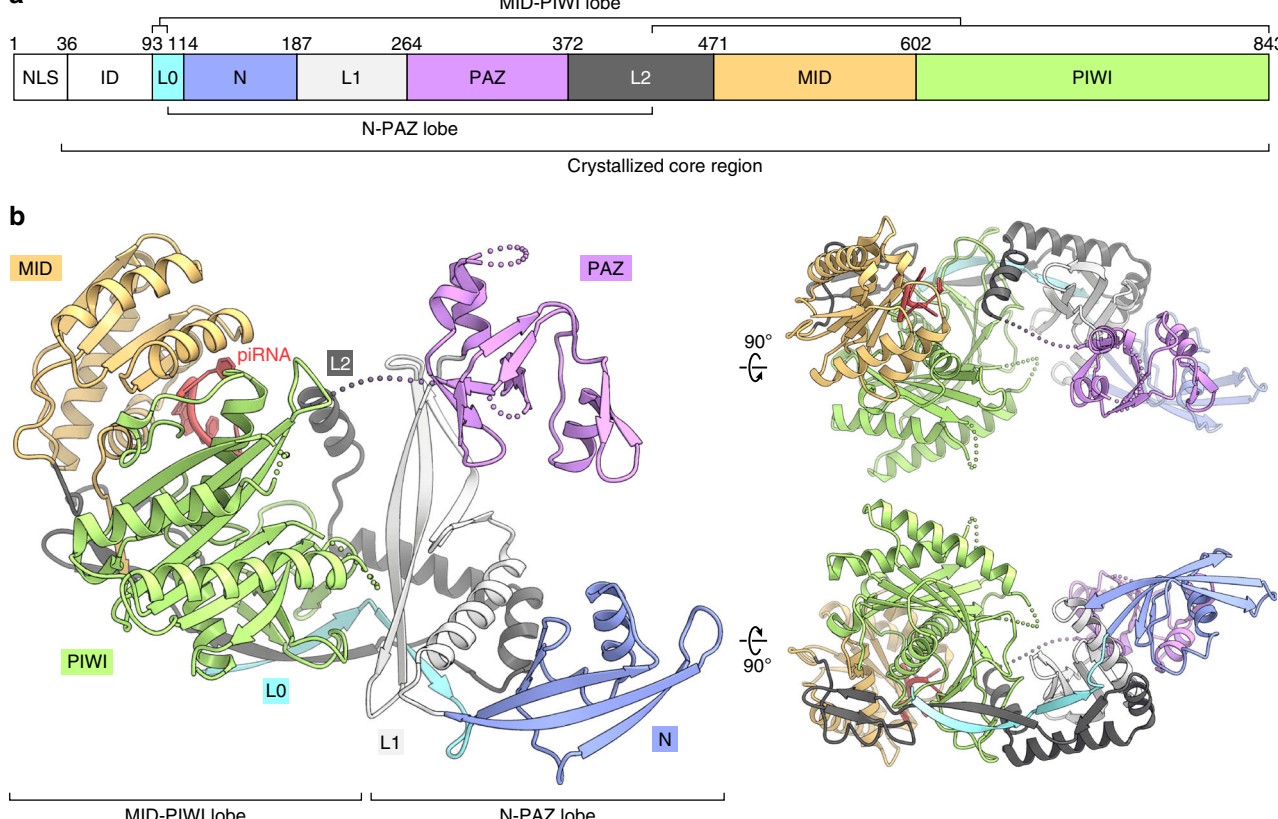

**Fig. 1 Overall structure. a** Domain organization of Piwi. NLS, nuclear localization signal; ID, intrinsically disordered region. **b** Crystal structure of the Piwi–piRNA complex.

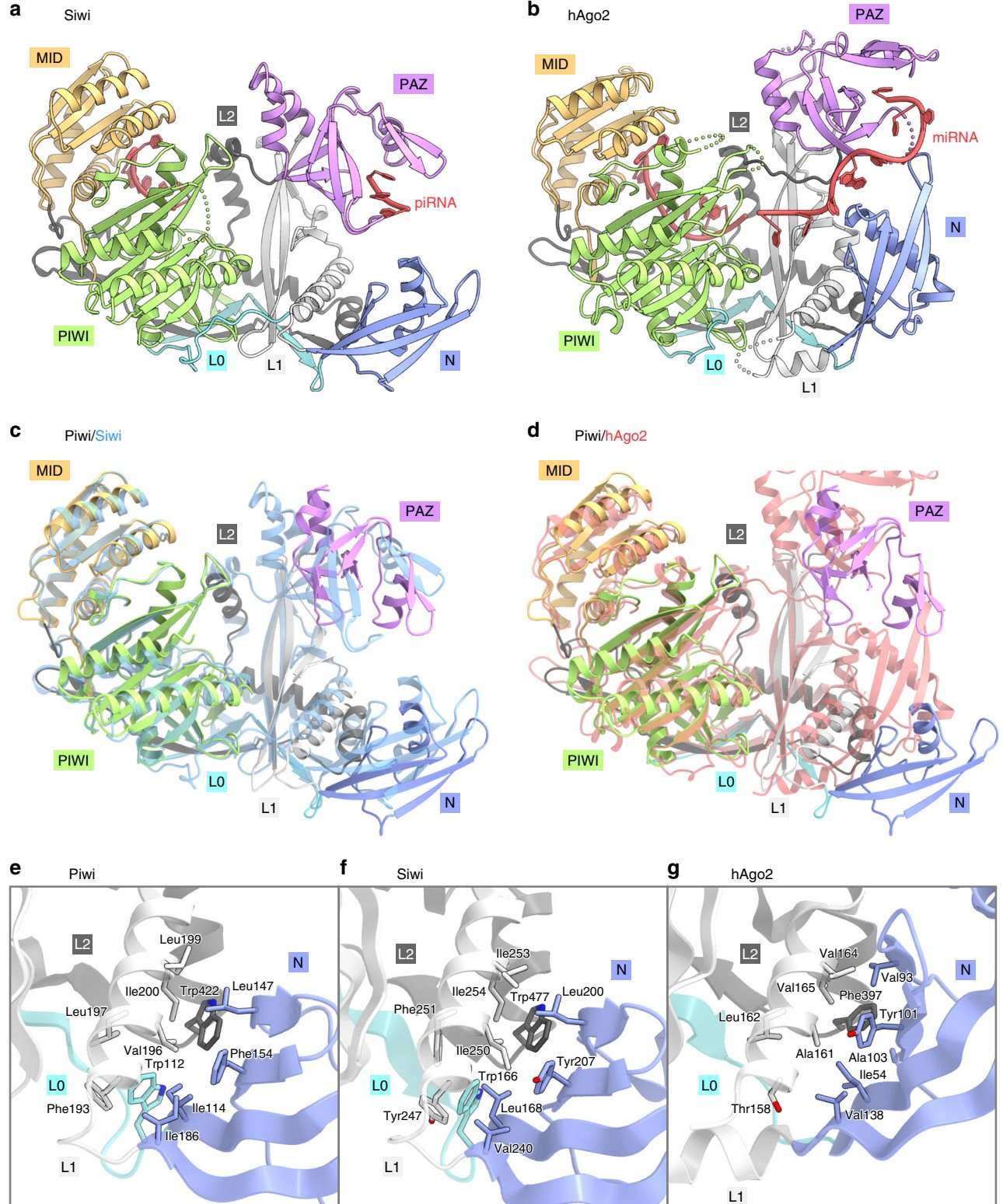

**Fig. 2 Structural comparison. a, b** Crystal structures of Siwi (PDB: 5GUH) **a** and hAgo2 (PDB: 4W5N) **b**. **c, d** Superimposition of Piwi on Siwi (blue) **c** and hAgo2 (red) **d**, based on their PIWI domains. Note that the Piwi PAZ domain is not well resolved in the density map, suggesting its flexibility. **e-g** N domain interfaces of Piwi **e**, Siwi **f**, and hAgo2 **g**.

play crucial roles in stabilizing the N domain arrangement (Fig. 2e, f). Trp112 (L0) and Trp422 (L2) of Piwi form a hydrophobic core with hydrophobic residues from the N domain (Ile114, Leu147, Phe154, and Ile186) and the L1 domain (Phe193, Val196, Leu197, Ile199, and Ile200) (Fig. 2e), as observed in the Siwi structure[48] (Fig. 2f). In hAgo2, Phe397, which is equivalent to Trp422 of Piwi, forms a hydrophobic core in a distinct manner, with residues from the N domain (Ile54, Val93, Tyr101, and Ala103) and the L1 domain (Ala161, Val164, and Val165)[34] (Fig. 2g). These structural observations indicated that the

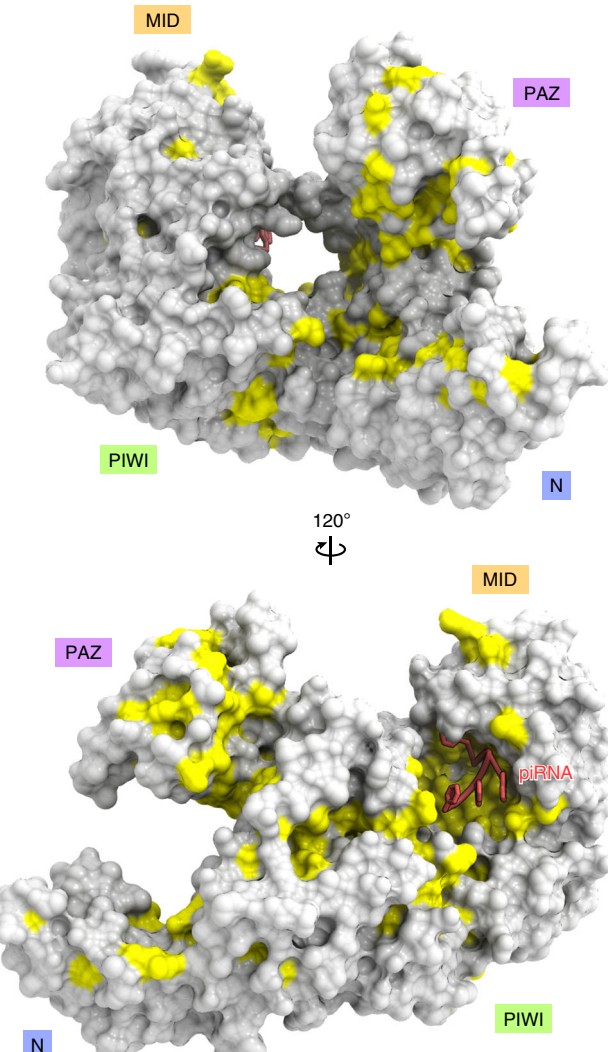

**Fig. 3 Surface conservation.** The residues conserved among Piwi, Aub, and Ago3 are colored yellow on the molecular surface of Piwi.

N domains of the PIWIs have similar arrangements, which are different from those of the AGOs. A mapping of the conserved residues among Piwi, Aub, and Ago3 revealed that the residues around the piRNA-binding region, but not the other regions, are highly conserved among the three PIWIs (Fig. 3), consistent with the fact that they function with distinct cofactors, such as Arx and Panx for Piwi[13,15,18,19].

**Recognition of the piRNA 5′ end.** The 5′ end of the bound piRNA is flipped into a pocket between the MID and PIWI domains of Piwi, whereas nucleotides 2–4 of the piRNA adopt an A-form-like conformation with their Watson–Crick edges exposed to the solvent (Fig. 4a), as in Siwi (Fig. 4b) and hAgo2 (Fig. 4c). The base of nucleotide 2 interacts with Ile582, and its phosphate is recognized by Thr570 and Thr573 (Fig. 4a). The phosphate groups of nucleotides 3 and 4 are recognized by Gln589 and Tyr801/Asn803, respectively (Fig. 4a).

The 5′ phosphate group of the piRNA is recognized by the side chains of Tyr551, Lys555, Gln567, and Lys593, and the main-chain amide group of Val568 (Fig. 4a). These residues are highly conserved among the PIWIs and AGOs (Supplementary Fig. 4). We observed an anomalous difference density in the vicinity of the 5′ phosphate group, where we modeled a zinc ion derived from the crystallization solution (Supplementary Fig. 5a). The

zinc ion is coordinated by phosphates 1 and 3 in the piRNA, the side chain of Gln589, and the C-terminal carboxyl group of Leu843 (Fig. 4a). Gln589 of Piwi is conserved in the PIWIs such as Siwi (Gln645) (Supplementary Fig. 4), and Siwi also recognizes the piRNA 5′ phosphate in a metal-dependent manner[48] (Fig. 4b). Similarly, the prokaryotic AGOs, such as TtAgo[38–41], RsAgo[42,43], MjAgo[46], and CbAgo[47], recognize the 5′ phosphate of the guide strand in a metal-dependent manner. In contrast, Gln589 of Piwi is replaced with a lysine residue in the eukaryotic AGOs, such as hAgo2 (Supplementary Fig. 4), in which Lys566 directly recognizes the 5′ phosphate of the guide RNA[33] (Fig. 4c). These observations reinforced the notion that, like the prokaryotic AGOs, the PIWIs recognize the piRNA 5′ phosphate in a metal-dependent manner.

The U1 nucleotide interacts with a loop region (residues 544–547; referred to as a specificity loop) in the MID domain of Piwi (Fig. 4a). The N3 of U1 form a hydrogen bond with the main-chain carbonyl group of Asn547, whereas its nucleobase forms a stacking interaction with the side chain of Tyr551 on the following α helix (Fig. 4a). The conformation of the specificity loop is stabilized by hydrogen bonds between Arg550 and Asp519/Asn545 and a van der Waals interaction between Arg550 and Pro544/Asn547 (Supplementary Fig. 5b). These structural features are consistent with the fact that Piwi prefers uridine at the piRNA 5′ end[7,49,50].

Siwi recognizes the U1 nucleotide via a hydrogen bond between the N3 of U1 and the main-chain carbonyl group of Tyr603, corresponding to Asn547 of Piwi[48] (Fig. 4b). Notably, the specificity loops of Piwi (544-PNDN-547) and Siwi (600-ARNY-603) consist of different residues (Supplementary Fig. 4), but adopt similar conformations, which are stabilized by distinct interactions (Supplementary Fig. 5b, c). In Siwi, Arg606 (Arg550 of Piwi) forms hydrogen bonds with Asp574/Asp605 (Asp519/Glu549 of Piwi), and the side chain of Arg606 is sandwiched by Pro576/Tyr603 (Thr521/Asn547 of Piwi) (Supplementary Fig. 5c). These structural findings explain why Piwi and Siwi prefer the U1 nucleotide, despite their different specificity-loop residues.

**Recognition of the piRNA 3′ end.** The piRNA 3′ end is disordered in the present structure, whereas the 2′-O-methylated piRNA 3′ end is recognized by the PAZ domain in the structures of Siwi[48], Hiwi1[52], and Miwi[53]. To examine whether the Piwi PAZ domain recognizes the piRNA 3′ end, we measured the binding of the isolated Piwi PAZ domain to an 8-mer RNA containing a 2′-O-methyl group at its 3′ end, using isothermal titration calorimetry (ITC). We found that the SUMO-tagged Piwi PAZ domain, but not the SUMO protein, binds the 8-mer RNA with a $K_d$ of 4.0 μM (Figs. 5a, b), which is comparable to the $K_d$ values for the PAZ domains of Hiwi1 (6.5 μM), Hiwi2 (2.0 μM), and Hili (10 μM)[52]. These results indicated that the Piwi PAZ domain recognizes the piRNA 3′ end, as in the other PIWI proteins. Consistent with this, the residues interacting with the piRNA 3′ end in the other PIWIs are conserved in Piwi (Supplementary Fig, 6).

**Piwi is not a slicer.** Most Argonaute proteins, such as KpAgo[29], hAgo2[32,33], and Siwi[48], have the DEDX (X is H or D) catalytic tetrads in their PIWI domains, which are responsible for the target RNA cleavage (Supplementary Fig. 4). The DEDH tetrads of hAgo2 and Siwi consist of Asp597/Glu637/Asp669/His807 and Asp670/Glu708/Asp740/His874, respectively (Fig. 6a, b). In the structures of hAgo2 and Siwi, the second Glu residues (referred to as a glutamate finger) are located on a flexible loop and adopt different conformations. In hAgo2, Glu637 adopts a "plugged-in" conformation and forms hydrogen bonds with His600 and

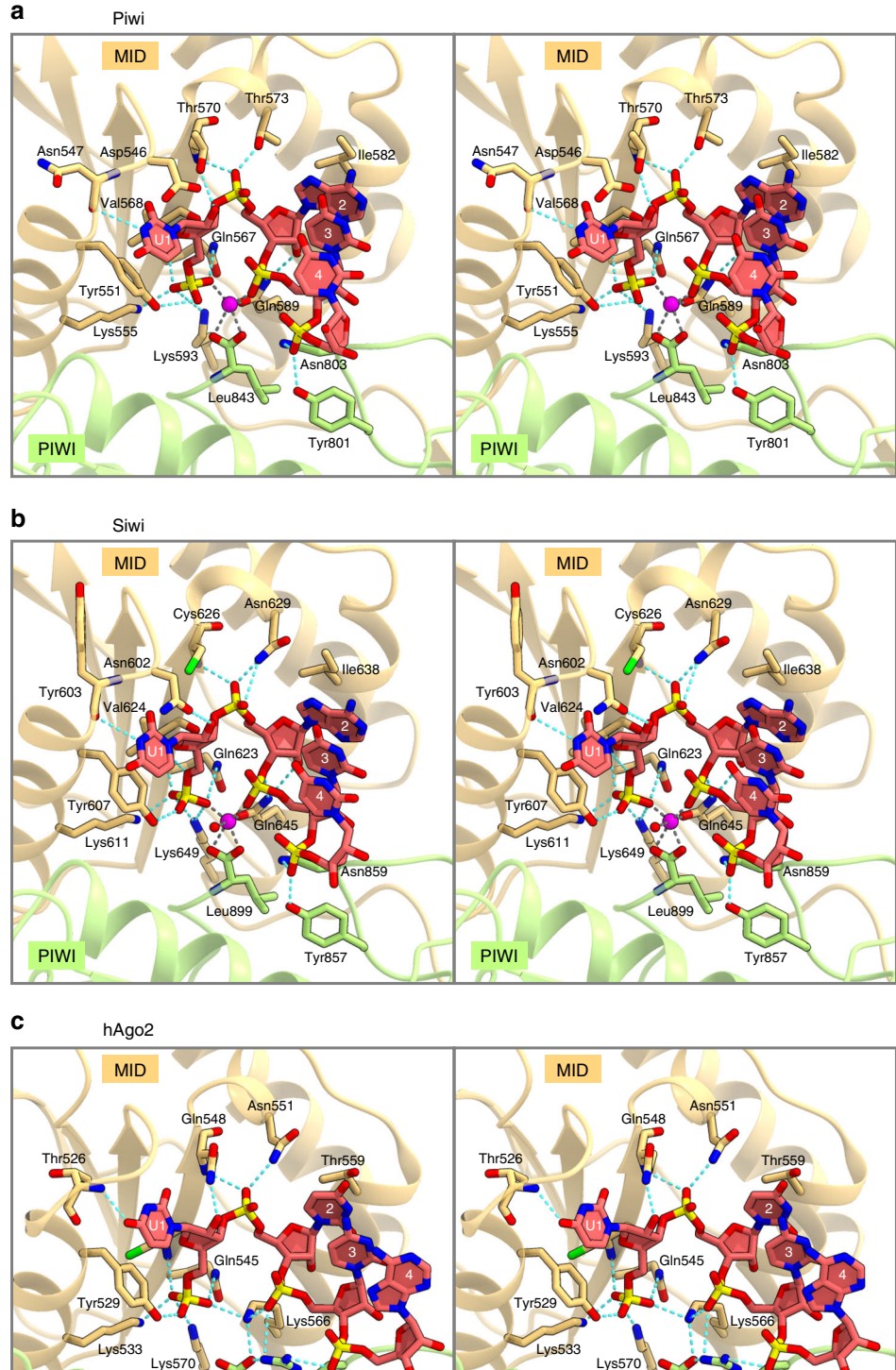

**Fig. 4 Recognition of the piRNA 5′ end. a–c** Guide RNA recognitions by Piwi **a**, Siwi (PDB: 5GUH) **b**, and hAgo2 (PDB: 4W5O) **c** (stereo view). The zinc and magnesium ions are shown as magenta spheres in **a** and **b**, respectively. Hydrogen bonds and electrostatic interactions are indicated by cyan dashed lines.

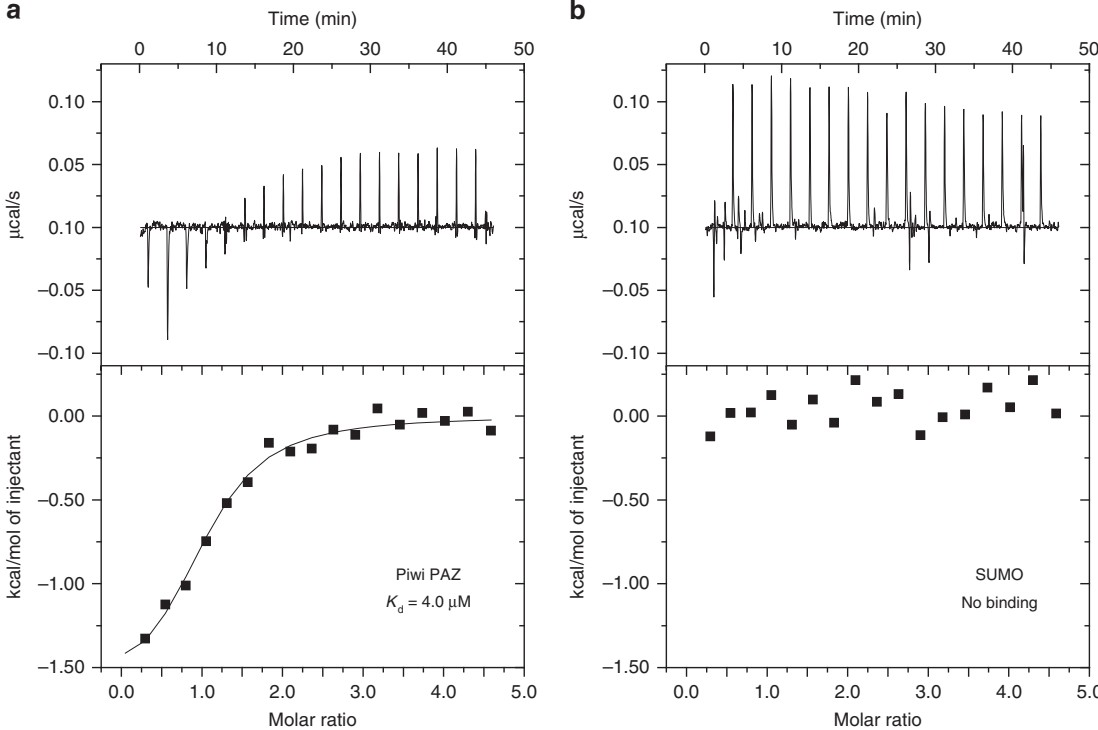

**Fig. 5 Recognition of the piRNA 3′ end. a, b** ITC experiments for the binding of the SUMO-tagged Piwi PAZ domain **a** or the SUMO protein **b** to the 8-mer RNA containing the 2′-O-methyl group at its 3′ end.

Arg668, which are in turn stabilized by hydrogen bonds with Ser610 and Glu683, respectively (Fig. 6a). In contrast, Glu708 is disordered and adopts an "unplugged" conformation in the Siwi structure (Fig. 6b). The prokaryotic AGOs, such as PfAgo and TtAgo, also have the DEDX catalytic tetrads. PfAgo has the DEDH tetrad (Asp558/Glu596/Asp628/His745), and Glu596 adopts the "unplugged" conformation in the apo structure[37] (Supplementary Fig. 7a). TtAgo has the DEDD tetrad (Asp478/Glu512/Asp546/Asp660), and Glu512 adopts the "unplugged" and "plugged-in" conformations in the guide-bound[39] and guide-target-bound[41] structures, respectively (Supplementary Fig. 7b, c).

To examine the importance of the catalytic tetrad for the RNA-guided RNA-cleaving "slicer" activity, we immunopurified the FLAG-tagged wild-type Siwi and the single (E708V and H874K) and double (E708V/H874K) mutants from BmN4 cells, and then measured their slicer activities toward a [32]P-labeled target RNA. As expected, the wild-type Siwi cleaved the target RNA (Fig. 6c). In contrast, the H874K and E708V/H874K mutants had almost no slicer activity (Fig. 6c), although they were loaded with piRNAs as in wild-type Siwi (Supplementary Fig. 8a). The E708V mutant showed reduced slicer activity, as compared with wild-type Siwi (Fig. 6c). These results indicated that His874 is essential for the slicer activity, with Glu708 facilitating the target cleavage.

The present structure revealed that Piwi has the DVDK (Asp614/Val653/Asp685/Lys818), rather than DEDH, tetrad in the PIWI domain, and that Val653, corresponding to the glutamate finger, adopts the "unplugged" conformation (Fig. 6d). In addition, His600 and Ser610 of hAgo2, which stabilize the "plugged-in" conformation of the glutamate finger (Glu637), are replaced with Lys617 and Ala625 in Piwi, respectively (Fig. 6d). To examine whether Piwi is a slicer, we immunopurified the FLAG-tagged Piwi from OSCs, and then measured its slicer activity toward a 72-nt target RNA (flam target), which is complementary to flam-piRNA-1, one of the major Piwi-bound piRNAs derived from the *flamenco* (*flam*) piRNA cluster in

OSCs[54] (Fig. 6e). The purified Piwi did not cleave the target RNA efficiently (Fig. 6f), although it was loaded with piRNAs (Supplementary Fig. 8b). These results indicated that, unlike hAgo2 and Siwi, Piwi is not a slicer.

**Slicer-Piwi cleaves target RNA**. We next examined whether the K617H/A625S/V653E/K818H (HSEH) mutant of Piwi, with a DEDH tetrad similar to those of hAgo2 and Siwi, exhibits the slicer activity. We immunopurified the FLAG-tagged HSEH mutant from OSCs, and then measured its slicer activity toward the flam target RNA. In contrast to wild-type Piwi, the HSEH mutant cleaved the flam target RNA, yielding 45–47-nt fragments (Fragment A) and a 25-nt fragment (Fragment A*) (Fig. 6f and Supplementary Fig. 8b). Fragment A appeared as a mixture of 45–47-nt RNAs, whereas Fragment A* appeared as a 25-nt single band (Fig. 6f), suggesting that the 45- and 46-nt by-products originate from the 70–71-nt flam targets lacking one or two nucleotides at the 3′ end. In addition to Fragments A and A*, we observed 53–55-nt fragments (Fragment B) and a 17-nt fragment (Fragment B*) (Fig. 6f). We assume that the flam target RNA was also cleaved by the HSEH mutant loaded with flam-piRNA-2, one of the other Piwi-bound piRNAs, yielding Fragments B and B* (Fig. 6e). The flam-piRNA-2 has 4-bp mismatches against the flam target RNA at its piRNA 3′ end (Fig. 6e), suggesting that a few mismatches at the piRNA 3′ end are tolerated for the target cleavage, as observed in the mouse PIWI protein Miwi[55]. Together, our structural and biochemical data revealed that Piwi has the non-canonical DVDK tetrad and lacks slicer activity, whereas the Piwi mutant with the reconstructed canonical DEDH tetrad (referred to as slicer-Piwi) can catalyze the piRNA-guided target RNA cleavage.

**Slicer-Piwi silences the *mdg1* transposon**. To examine whether the Piwi HSEH mutant (slicer-Piwi) can repress transposons, we

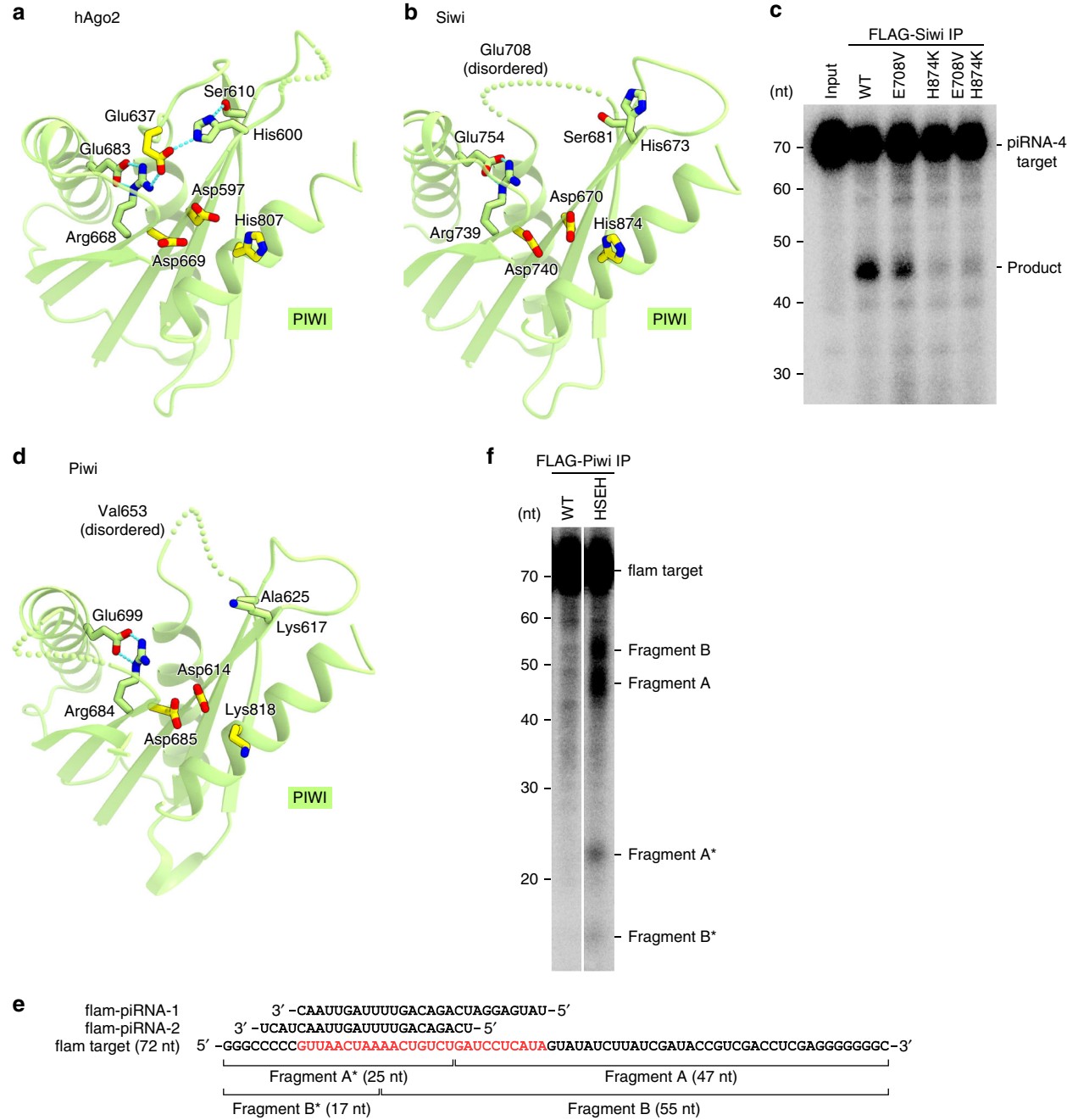

**Fig. 6 Catalytic tetrad. a, b** DEDH tetrads of hAgo2 (PDB: 4W5N) **a** and Siwi (PDB: 5GUH) **b. c** Slicer activities of Siwi. The immunopurified FLAG-tagged wild-type Siwi and mutants were incubated with the internally $^{32}$P-labeled substrate RNA (piRNA-4 target), and the reaction products were analyzed by denaturing urea-PAGE. **d** DVDK tetrad of Piwi. **e** Sequences of the target RNA (flam target) and the piRNAs (flam-piRNA-1 and flam-piRNA-2). **f** Slicer activities of Piwi. The immunopurified FLAG-tagged wild-type Piwi or slicer-Piwi was incubated with the internally $^{32}$P-labeled substrate RNA (flam target), and the reaction products were analyzed by denaturing urea-PAGE. HSEH, the slicer-Piwi K617H/A625S/V653E/K818H mutant. Source data are provided as a Source Data file.

expressed either the FLAG-tagged wild-type Piwi or slicer-Piwi in Piwi-depleted OSCs, and then monitored the expression levels of the *mdg1* transposon. We confirmed that wild-type Piwi and slicer-Piwi were expressed at comparable levels (Supplementary Fig. 8c, d). Notably, slicer-Piwi rescued the defect in transposon silencing caused by the loss of endogenous Piwi, as efficiently as wild-type Piwi (Fig. 7a), indicating that slicer-Piwi can silence the *mdg1* transposon in OSCs.

Given that slicer-Piwi cleaves the target RNAs, it is possible that, like Aub and Ago3, slicer-Piwi silences transposons at post-transcriptional, rather than transcriptional, levels. To explore the silencing mechanism of slicer-Piwi, we expressed either the FLAG-tagged wild-type Piwi or slicer-Piwi in OSCs, and then examined their interactions with Arx, an essential cofactor for Piwi-mediated transcriptional silencing[13,15]. The wild-type Piwi and slicer-Piwi were co-purified with Arx (Fig. 7b), and slicer-Piwi failed to silence transposons in Arx-depleted OSCs (Fig. 7c and Supplementary Fig. 8e). These results indicated that, like wild-type Piwi, slicer-Piwi co-transcriptionally silences the *mdg1* transposon.

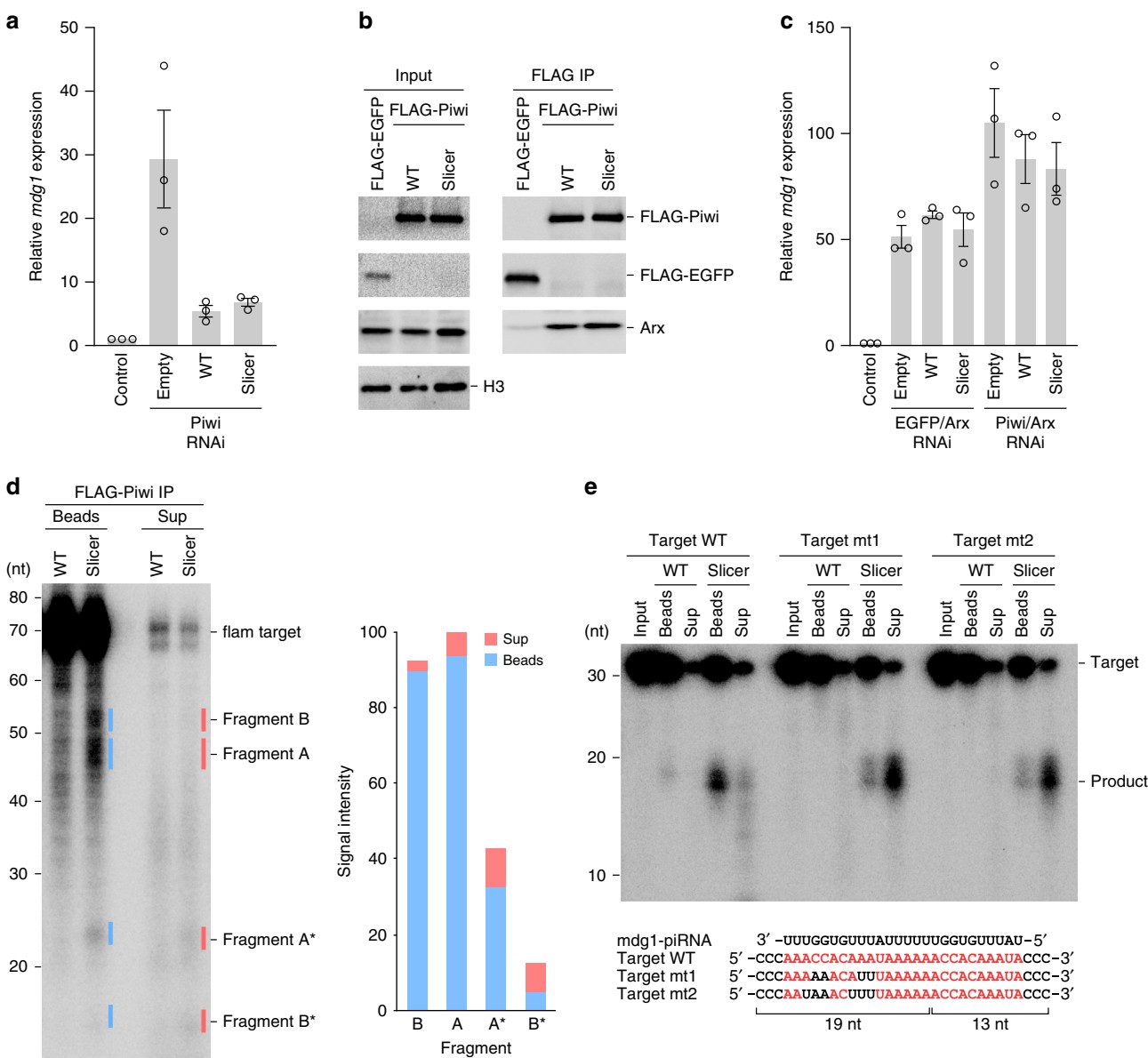

**Fig. 7 Transposon silencing. a** Piwi-mediated silencing of the *mdg1* transposon. FLAG-tagged wild-type Piwi or slicer-Piwi was expressed in endogenous Piwi-depleted OSCs, and the expression levels of the *mdg1* transposon were examined by quantitative RT-PCR ($n = 3$; error bars indicate SEM). Empty, empty vector control; Slicer, the slicer-Piwi K617H/A625S/V653E/K818H mutant. **b** Binding of Piwi to Arx. FLAG-tagged wild-type Piwi or slicer-Piwi was expressed in endogenous Piwi-depleted OSCs, and the proteins were then immunoprecipitated using anti-FLAG beads. The cell lysates and immunoprecipitates were analyzed by western blotting, using the indicated antibodies. H3 was used as a loading control. **c** Requirement of Arx for Piwi-mediated silencing. FLAG-tagged wild-type Piwi or slicer-Piwi was expressed in endogenous Piwi/Arx-depleted OSCs, and the expression levels of the *mdg1* transposon were examined by quantitative RT-PCR ($n = 3$; error bars indicate SEM). **d** Binding of slicer-Piwi to the cleavage products. FLAG-tagged wild-type Piwi or slicer-Piwi was immunopurified from OSCs, and the proteins were then incubated with the internally [32]P-labeled substrate RNA (flam target). The supernatant and beads fractions were analyzed by denaturing urea-PAGE. Signal intensities of the fragments are shown on the right. **e** Binding of Piwi to partially complementary targets. The immunopurified FLAG-tagged wild-type Piwi and slicer-Piwi were incubated with the 5′ [32]P-labeled substrate RNA, and the supernatant and beads fractions were then analyzed by denaturing urea-PAGE. Sequences of the target RNAs (WT, mt1, and mt2) and the piRNA (mdg1-piRNA) are shown below. Source data are provided as a Source Data file.

**Slicer-Piwi dissociates from partially complementary targets**. A previous study showed that Siwi remains bound to the cleaved target RNAs, and requires the RNA helicase Vasa to release the cleavage products, thereby facilitating piRNA amplification[56]. To examine whether slicer-Piwi remains bound to the cleavage products, we incubated the flam target RNA with the FLAG-tagged slicer-Piwi, purified from OSCs using anti-FLAG beads, and then analyzed the RNAs in the supernatant and beads fractions. Most of the cleavage products were detected in the beads

fraction, rather than the supernatant (Fig. 7d), indicating that the cleaved RNAs are primarily bound to slicer-Piwi. These results suggested that slicer-Piwi does not release its cleavage products autonomously, as in the case of Siwi[56]. Given that Piwi is unlikely to have a partner RNA helicase in the nucleus, slicer-Piwi probably remains associated with the *mdg1* transposon transcript even after the cleavage, thereby resulting in co-transcriptional silencing in OSCs. We also detected small amounts of the cleavage products, particularly the shorter Fragments A* and B*, in

the supernatant fraction (Fig. 7d), suggesting that some of the cleavage products can be released from slicer-Piwi. Notably, slicer-Piwi more efficiently released the cleavage products that are less complementary to the mdg1-piRNA, whereas wild-type Piwi tightly associated with the three targets regardless of the degree of their complementarity (Fig. 7e). These results revealed that, as compared with slicer-deficient wild-type Piwi, slicer-Piwi more readily dissociates from its target RNAs, particularly less-complementary targets.

## Discussion

Piwi is the founding member of the PIWI-clade Argonautes, and *piwi* (P-element induced wimpy testis) was identified in 1997 as a critical gene for germline stem cell division[57]. Subsequently, a series of studies showed that (1) *piwi* is involved in transposon silencing[58], (2) Piwi binds piRNAs[7,8,50,59], and (3) Piwi co-transcriptionally silences transposons via heterochromatin formation[9–12]. Despite its functional importance, no structural information has been reported for Piwi for over 20 years since its discovery. In this study, we determined the crystal structure of the Piwi–piRNA complex, which represents the second example of the PIWI-clade Argonaute structures. A structural comparison between Piwi and Siwi highlights the PIWI-specific structural features, such as the overall domain arrangement and the metal-dependent piRNA recognition.

Our structural and functional data revealed that Piwi has the non-canonical DVDK tetrad in the PIWI domain and the FLAG-tagged Piwi purified from OSCs does not cleave the target RNA efficiently, thereby establishing that Piwi is not a slicer. We previously reported that Piwi exhibited the slicer activity in vitro[50]. Given that an excess amount (~ 1 µg) of the GST-tagged Piwi purified from *Escherichia coli* was used for in vitro cleavage experiments[50], it is likely that we detected negligible, if any, slicer activity by Piwi in the previous study.

Our ITC experiments indicated that the PAZ domain of Piwi recognizes the piRNA 3′ end, as observed in the PAZ domains of the other PIWI proteins. Recently, we reported that the Piwi PAZ mutant (Y327A/Y328A) associates with less mature piRNAs in OSCs, highlighting the importance of the PAZ-mediated piRNA recognition for the piRNA maturation[60]. The present structure suggested that the PAZ domain of Piwi has conformational flexibility. Notably, the length distribution of the Piwi-bound piRNAs (~ 23–30 nt with the peak of 26 nt)[49] is wider than that of the Siwi-bound piRNAs (~ 27–29 nt with the peak of 28 nt)[56]. These observations suggest that Piwi can accommodate piRNAs of different lengths, owing to the flexibility of its PAZ domain.

The Piwi–piRNA complex associates with target transposon transcripts and the Panx-Nxf2-Nxt1 complex, thereby recruiting the chromatin silencing machinery to target transposon loci[18–23]. We found that Piwi is not a slicer and stably associates with the less-complementary targets, whereas the slicer-Piwi mutant dissociates from the less-complementary targets after slicer-mediated cleavage. These observations suggest that, if Piwi were an active slicer, then the Piwi–piRNA complex could dissociate from partially complementary transposon transcripts after slicer-mediated RNA cleavage. Thus, the slicer activity likely compromises Piwi-mediated co-transcriptional silencing, although slicer-Piwi silenced the fully complementary *mdg1* transposon in OSCs. Taken together, we propose that Piwi lost the slicer activity during its molecular evolution to serve as a piRNA-guided RNA-binding platform, thereby ensuring faithful co-transcriptional silencing of transposons.

Siwi remains bound to the cleavage products and requires the RNA helicase Vasa for the product release[56], whereas the eukaryotic AGOs readily dissociate from the cleavage products[56,61–63]. We found that slicer-Piwi remains bound to the cleavage products, and the N domains of Piwi and Siwi adopt similar arrangements, which are different from that of hAgo2. In addition, the N domain arrangements of Piwi and Siwi are relatively similar to that of the prokaryotic MpAgo, in which the N domain interacts with a fully paired guide-target duplex[45]. These observations suggest that the N domains of the PIWIs contribute to the stable association with the guide-target duplex, although the PIWI structure bound to a defined piRNA and its target RNA is required to elucidate its target recognition mechanism.

In summary, this study provides a basis toward a mechanistic understanding of Piwi-mediated transposon silencing. Structural elucidation of the Piwi–piRNA complex bound to cofactors, such as Arx and Panx, will be required to understand how the Piwi–piRNA complex and the cofactors cooperate to co-transcriptionally silence transposons. In addition, a functional analysis of slicer-Piwi in a fly model will be important to clarify the effects of the slicer activity on the co-transcriptional silencing.

## Methods

**Cell lines**. OSCs were obtained from fGS/OSS[49], and cultured at 27 °C in Shields & Sang M3 Insect Medium (Sigma-Aldrich), supplemented with 10% fly extract[49], 10% fetal bovine serum (Funakoshi), 1% L-glutathione reduced (Sigma-Aldrich), and 1% human recombinant insulin (Wako).

**Purification**. The anti-Piwi monoclonal antibody[50] (100 mg) (Mikuri Immunology Laboratory) was coupled with CNBr-activated Sepharose 4 Fast Flow beads (10 mg) (GE Healthcare), according to the manufacturer's instructions. OSCs (~ 5.0 × 10$^{10}$ cells) were suspended in buffer (30 mM Tris-HCl (pH 7.3), 300 mM NaCl, 1 mM EDTA, 1 mM DTT, 10% glycerol, 0.1% NP-40, 4.5 µg/ml aprotinin, 1.4 µg/ml leupeptin, and 2.0 µg/ml pepstatin), lysed by homogenization, and then centrifuged. The supernatant was incubated with anti-Piwi antibody-coupled beads at 4 °C for 3.5 h in an Econo-Column (Bio-Rad). The beads were washed with buffer (50 mM Tris-HCl (pH 8.0), 300 mM NaCl, 0.5 mM CaCl$_2$, 1 mM DTT, 10% glycerol, and 0.1% NP-40), and then incubated with chymotrypsin (28 µg) (Promega) at 4 °C for 18 h, to release the Piwi–piRNA complex. The Piwi–piRNA complex was loaded onto a HiTrap Heparin HP column (GE Healthcare), equilibrated with buffer (50 mM Tris-HCl (pH 8.0), 200 mM NaCl, 1 mM DTT, and 10% glycerol), and eluted using a linear gradient of 0.2–2 M NaCl. The Piwi–piRNA complex was further purified by chromatography on a HiLoad 16/600 Superdex 200 column (GE Healthcare), equilibrated with buffer (10 mM Tris-HCl (pH 8.0), 300 mM NaCl, 1 mM DTT, and 10% glycerol).

For ITC experiments, the His-SUMO-tagged PAZ domain of Piwi (residues 262–374) was expressed in *E. coli* Rosetta 2 (DE3) (Novagen) and purified by chromatography on Ni-NTA Superflow (QIAGEN) and HiLoad 16/600 Superdex 75 (GE Healthcare) columns, equilibrated with buffer (10 mM Tris-HCl (pH 8.0) and 500 mM NaCl). The His-tagged SUMO protein was prepared using a protocol similar to that used for the His-SUMO-tagged PAZ domain. As the isolated Piwi PAZ domain precipitated when the SUMO tag was removed by the SUMO protease treatment, the His-SUMO-tagged PAZ domain was used for ITC experiments.

**Crystallization**. The purified Piwi–piRNA complex was crystallized at 20 °C, using the sitting-drop vapor diffusion method. The crystallization drops were formed by mixing 0.2 µl of Piwi–piRNA solution ($A_{280nm}$ = 11) and 0.2 µl of reservoir solution (100 mM MES-NaOH (pH 6.0), 10 mM ZnCl$_2$, 20% PEG 6000). Since the resolution was improved by the addition of CH$_3$HgCl, the crystals were incubated in the reservoir solution supplemented with 35% ethylene glycol and 0.1–100 mM CH$_3$HgCl at 20 °C for 3 h, and were then flash-cooled in liquid nitrogen for data collection.

**Data collection and structure determination**. X-ray diffraction images were collected using an EIGER X 9 M detector at a wavelength of 1.000 Å on BL32XU, SPring-8. From each crystal, 180° data were collected using a helical data collection scheme. The 32 diffraction data sets were indexed and integrated with the KAMO pipeline[64], using DIALS[65] with the scan_varying option. Integrated intensities were then hierarchically clustered using normalized structure factor amplitudes, and merged using XSCALE[66] with outlier rejections implemented in KAMO. Finally, 23 diffraction data sets were merged into a high-quality data set.

The structure was solved by molecular replacement with Phaser[67], using the Siwi model (PDB: 5GUH), which was modified using Sculptor[68]. The PAZ domain was not fitted to the electron density map, and a weak density blob was observed at a displaced position. The PAZ domains of Piwi and Siwi share 40% sequence identity, indicating that their secondary structures are similar. Indeed, a Siwi-based homology model (100% confidence) of the Piwi PAZ domain was generated with

the Phyre2 server[69]. The homology model was manually placed at the center of the blob, and then the orientation and translation were optimized against the $2mF_O - DF_C$ map, using the constrained real space search[70]. The structural model was manually modified using Coot[71], and refined using Refmac5[72], with the SAD function[73] and the Siwi-based external restraints prepared with Prosmart[74]. The structural model was further refined using phenix.refine[75], with secondary-structure restraints. In the final model, the PAZ domain was well fitted to the $mF_O - DF_C$ omit map, with the real space correlation coefficient of 0.6. The $R_{free}$ value was decreased from 0.2627 to 0.2585 by the inclusion of the PAZ domain. Thus, the PAZ domain was included into the final model, although it is not perfectly resolved in the electron density map, probably owing to its flexibility. The model building was aided by the anomalous difference densities of mercury ions bound to Cys188, Gln243, Cys271, Cys317, Gln512, Cys531 Ser534, Cys648, Asp685, and Cys814, and zinc ions bound to U1, U3, His116, His118, Gln152, His244, His446, Asp473, Asp476, His488, Gln589, Asp614, Asp685, His829, and Leu843.

**ITC**. Binding of the 8-mer RNA containing the 2′-O-methyl group at its 3′ end (ACCGACUU$_m$) to the SUMO-tagged Piwi PAZ domain and the SUMO protein (as a control) was measured at 20 °C, using a MicroCal iTC200 (GE Healthcare). The 8-mer RNA was purchased from GeneDesign, and dissolved in the gel filtration buffer (10 mM Tris-HCl (pH 8.0) and 500 mM NaCl). The 8-mer RNA (0.50 mM) was injected 18 times (0.4 μl for injection 1 and 2 μl for injections 2–18) into the protein solution (20 μM SUMO-PAZ or 20 μM SUMO in the gel filtration buffer), with 150 s intervals between injections. The concentrations of the protein and RNA samples were determined using the BCA protein assay kit (TaKaRa) and the absorbance at 260 nm, respectively. Data were analyzed using the Origin7 software (MicroCal). Data obtained from injections into the buffer were subtracted from the sample data before data analysis. Measurements were repeated at least twice, and similar results were obtained.

**Small RNA isolation**. Immunoprecipitation and small RNA isolation were performed as described previously[48]. In brief, OSCs or BmN4 cells ware lysed in buffer (30 mM HEPES-KOH (pH 7.4), 500 mM NaCl, 150 mM KOAc, 5 mM Mg(OAc)$_2$, 5 mM DTT, 0.1% NP-40, 2 μg/ml pepstatin, 2 μg/ml leupeptin, and 0.5% aprotinin (Wako)), and then centrifuged. The supernatants were incubated with Dynabeads Protein G (Thermo Fisher Scientific) bound to an anti-DDDDK-tag mAb (MBL, FLA-1). The beads were washed twice with the buffer and twice with the buffer without 500 mM NaCl, and then treated with Proteinase K and phenol-chloroform. The liberated RNAs were precipitated with ethanol, dephosphorylated with Antarctic Phosphatase (NEB), and then radiolabeled with $^{32}$P-γ-ATP (PerkinElmer) and T4 PNK (NEB).

**Plasmid construction**. The slicer-Piwi mutant and the Siwi mutants were generated by inverse PCR, using pAcF-Piwi[51] and the FLAG-Siwi vector[48] as the templates, respectively. The gene encoding the Piwi PAZ domain (residues 262–374) was amplified by PCR using pAcF-Piwi[51] as the template, and then cloned into the pE-SUMO vector (LifeSensors). The sequences of the DNA oligos used for PCR are listed in Supplementary Table 1.

**In vitro RNA cleavage assay**. In vitro cleavage assays were performed as described previously[56]. FLAG-tagged wild-type Siwi and the FLAG-tagged Siwi mutants were expressed in BmN4 cells, and the proteins were then immunopurified using Dynabeads Protein G and the anti-DDDDK-tag mAb. The purified proteins were incubated at 27 °C for 17 h with the internally $^{32}$P-labeled substrate RNA (piRNA-4 target)[56], and the reaction products were then analyzed by denaturing urea-PAGE. FLAG-tagged wild-type Piwi or slicer-Piwi was expressed in OSCs, and the proteins were then immunopurified using Dynabeads Protein G and the anti-DDDDK-tag mAb. The purified proteins were incubated at 27 °C for 17 h with the internally $^{32}$P-labeled flam target RNA or the 5′ $^{32}$P-labeled mdg1 target RNA, and the reaction products were then analyzed by denaturing urea-PAGE. The flam target RNA was transcribed in vitro with a T7 High Yield Transcription kit (Epicenter), using $^{32}$P-α-UTP (PerkinElmer). The template for the in vitro transcription was prepared by PCR using DNA oligos (Supplementary Table 1). The mdg1 target RNAs were purchased from Integrated DNA Technologies.

**Rescue assay**. Rescue assays were performed essentially as described previously[76]. In brief, FLAG-tagged wild-type Piwi or slicer-Piwi was expressed in OSCs, in which endogenous Piwi or Piwi/Arx was depleted by RNAi. OSCs were transfected with 600 pmol of siRNA duplex and 6 μg plasmid, using a Nucleofector device (Amaxa Biosystems). After transfection, the cells were incubated at 27 °C for 48 h, and the expression levels of the mdg1 transposon were then examined by quantitative PCR with reverse transcription (RT-PCR). For western blotting, anti-Arx[21] (hybridoma supernatant), anti-DDDDK-tag (1:10,000), anti-H3 (Abcam, ab1791) (1:2000), anti-Piwi[50] (1:1000), and anti-beta-tubulin E7 (DSHB) (1:1000) antibodies were used.

**Immunofluorescence**. Immunofluorescence was performed essentially as described previously[49]. In brief, OSCs were adhered to a cover glass coated with poly-L-lysine, and then fixed with 4% formaldehyde for 15 min at room temperature. After fixing, the cells were permeabilized with 0.1% Triton X-100 for 15 min at room temperature, and then stained with anti-FLAG M2 (Sigma-Aldrich, F3165, 1:1000 dilution) and Alexa Flour 546 goat anti-mouse (Thermo Fisher Scientific, A11030, 1:1000 dilution), as primary and secondary antibodies, respectively.

**Reporting summary**. Further information on research design is available in the Nature Research Reporting Summary linked to this article.

## Data availability

The atomic coordinates of the Piwi–piRNA complex have been deposited in the Protein Data Bank, with the accession number PDB: 6KR6. The X-ray diffraction images are available at the Zenodo data repository (https://doi.org/10.5281/zenodo.3603539). The source data underlying Figs. 6c, 6f, and 7a–e and Supplementary Figs 1b, 1d, 1e, 8a, 8b, 8d, and 8e are provided as a Source Data file. Other data are available from the corresponding authors upon reasonable request.

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

## Acknowledgements

We thank Dr. Masahiro Fukuda, Dr. Wataru Shihoya, and the beamline scientists at SPring-8 BL32XU for assistance with data collection, Dr. Takanori Nakane for assistance with structure determination, and Atsuhiro Tomita for assistance with the ITC experiments. This work was supported by JSPS KAKENHI Grant Numbers 17H05592 (H.N.), 18H02384 (H.N.), and 19H05466 (M.C.S.).

## Author contributions

S.Y. performed the sample preparation, structural analysis, and ITC experiments; A.O. performed the sample preparation and crystallization with assistance from S.H. and N.M.; K.M.N. and A.K. performed the functional analysis; K.Y. performed the structural analysis; N.D. performed the mass spectrometric analysis; R.I. assisted with the structural analysis; K.S. and H.S. assisted with the functional analysis; H.N. analyzed the data and wrote the manuscript with assistance from S.Y. and K.M.N.; H.N., M.C.S. and O.N. supervised all of the research.

## Competing interests

The authors declare no competing interests.
