## [Peer Review File · Nature Communications]

Reviewers' comments:

Reviewer #1 (Remarks to the Author):

In their manuscript 'Crystal structure of Drosophila Piwi' Yamaguchi et al. present the crystal structure of the Drosophila Argonaute protein Piwi in complex with endogenous piRNA. In many aspects, Piwi structurally resembles Siwi (silkworm Aubergine), whose atomic structure has been solved recently by the same authors. Besides an overall description of the structure, the authors find that Piwi (in contrast to Siwi) does not harbor a canonical catalytic tetrad required for target RNA cleavage and thus lacks slicing activity. By restoring a canonical catalytic tetrad the authors engineer a 'slicer-active Piwi' which is able to cleave target RNA. This slicer-Piwi is able to rescue a Piwi knockdown in OSCs, and the authors suggest that it is still able to initiate co-transcriptional silencing. The authors find that targets with limited number of mismatches to the piRNA display a faster displacement from slicer-Piwi compared to wildtype Piwi. Based on this the authors speculate that Piwi lost its slicer activity in order to increase its on-target time, which should facilitate more potent co-transcriptional silencing.

Overall the Piwi structure is very similar to the structure of silkworm Siwi, published previously by the same group. While the Drosophila Piwi structure is certainly an important aspect for the field, the manuscript in its current version provides limited novel insights for the piRNA biology field and does not open up a new domain of research. In light of this, the authors might want to address the following points.

1. The largely disordered PAZ domain

The most striking difference to the previously published Siwi structure is the disordered PAZ domain. The authors should investigate whether this could have a biological implication. Is the 3' end of the bound piRNA associated with the PAZ domain, as seen in other Argonaute structures? If yes, how can the PAZ domain still be largely disordered? If not, could the authors try to demonstrate that biochemically and discuss the implications?

I am further not convinced that the structural data allows the authors to place the PAZ domain with confidence. The localization between two putative mercury ions can hardly be accurate. The authors report a decrease in Rwork from 0.253 to 0.244, a rather modest improvement and only meaningful if the Rfree does significantly improve as well. Figure 2C suggest for the reader that there is a drastically different conformation of the PAZ domain in Siwi and Piwi and this is not supported sufficiently through the authors data. I suggest to not build the Paz domain in the model and to indicate its position in Figs 1 and 2 with a schematic representation.

2. Phenotype of 'slicer-Piwi'

The authors need to comment on their earlier findings (2006 GenesDev) that recombinant Drosophila Piwi is an active slicer. Despite this earlier report, other authors have suggested previously that Piwi would not have any RNA cleavage activity (based on the imperfect catalytic tetrad). The crystal structure together with biochemical analysis presented here proves that nicely. The fact that the authors are able to generate 'slicer-Piwi' through the mutation of key residues demonstrates that the inactivity of Piwi indeed is based on the altered catalytic tetrad and its environment. This is an interesting experiment.

In their rescue assay in OSC cells, the authors demonstrate that 'slicer-Piwi' does complement a Piwi knockdown and that 'slicer-Piwi' is able to initiate co-transcriptional transposon silencing (this is shown only indirectly as silencing is still Gtsf1 dependent). Based on in vitro experiments using target RNA that has impaired complementarity, the authors state that "slicing activity could compromise the Piwi-mediated co-transcriptional silencing" (lines 97-98). To strengthen their case, the authors should consider analyzing the effect of 'slicer-Piwi' in a fly model, rather than relying on the fairly rough rescue assay in cell culture. Only a fly rescue experiment will allow a strong statement about whether or not slicing does affect Piwi function.

Reviewer #2 (Remarks to the Author):

Piwi protein, identified in 1997 as a critical gene for germline stem cell division, binds piRNA to form piRNA-induced silencing complexes that silence transposons and maintain genome integrity. PIWI and AGO are two clades of Argonaute proteins. The structures of AGO proteins have been well studied in the last 15 years. However, the structural information of Piwi protein is very limited due to difficulties in sample preparation. To date, only one Piwi protein Siwi structure has been reported.

Yamaguchi et al. solved the crystal structure of *Drosophila* Piwi protein in complex with endogenous piRNAs. This is the second structure of the Piwi protein. Although the overall structure of Piwi is similar to Siwi protein, the structure of the Piwi-piRNA complex reveals some PIWI specific structural features including the overall domain arrangement and metal-dependent piRNA binding. Authors find that the Piwi protein has a non-canonical DVDK tetrad and lacks the slicer activity, but the DEDH mutant has the slicer activity. The authors propose that Piwi protein lack of slicer activity serves as a piRNA-guided RNA binding platform, ensuring faithful co-transcriptional silencing of transposons. Collectively, this study provides significant insights into the molecular mechanism of the Piwi-mediated transposon silencing.

Piwi proteins and prokaryotic Ago proteins share several structural features. 1. The binding of the 5'-phosphate of piRNA and siRNA are both metal-dependent. 2. The Glu finger from both the Piwi and prokaryotic Ago proteins are located away from the other three catalytic residues. Authors compared the structures of Piwi, Siwi, and hAgo2, but comparisons between Piwi and pfAgo and TtAgo are limited. Additional comparisons between the Piwi and pf/TtAgo are suggested.

In page 8 lines 215 and 216, Fig. 5A should be Fig. 5B, and Fig. 5B should be fig. 5A.

Figures 2C, 2D and S4 should be optimized to improve their clarity.

Reviewer #1 (Remarks to the Author):

In their manuscript 'Crystal structure of Drosophila Piwi' Yamaguchi et al. present the crystal structure of the Drosophila Argonaute protein Piwi in complex with endogenous piRNA. In many aspects, Piwi structurally resembles Siwi (silkworm Aubergine), whose atomic structure has been solved recently by the same authors. Besides an overall description of the structure, the authors find that Piwi (in contrast to Siwi) does not harbor a canonical catalytic tetrad required for target RNA cleavage and thus lacks slicing activity. By restoring a canonical catalytic tetrad the authors engineer a 'slicer-active Piwi' which is able to cleave target RNA. This slicer-Piwi is able to rescue a Piwi knockdown in OSCs, and the authors suggest that it is still able to initiate co-transcriptional silencing. The authors find that targets with limited number of mismatches to the piRNA display a faster displacement from slicer-Piwi compared to wildtype Piwi. Based on this the authors speculate that Piwi lost its slicer activity in order to increase its on-target time, which should facilitate more potent co-transcriptional silencing.

Overall the Piwi structure is very similar to the structure of silkworm Siwi, published previously by the same group. While the Drosophila Piwi structure is certainly an important aspect for the field, the manuscript in its current version provides limited novel insights for the piRNA biology field and does not open up a new domain of research. In light of this, the authors might want to address the following points.

We thank the reviewer for appreciating the importance of the *Drosophila* Piwi structure. We have addressed the points raised by the reviewer as follows.

1. The largely disordered PAZ domain

The most striking difference to the previously published Siwi structure is the disordered PAZ domain. The authors should investigate whether this could have a biological implication. Is the 3' end of the bound piRNA associated with the PAZ domain, as seen in other Argonaute structures? If yes, how can the PAZ domain still be largely disordered? If not, could the authors try to demonstrate that biochemically and discuss the implications?

We appreciate the helpful comments. According to the reviewer's comment, we examined whether the isolated PAZ domain of Piwi binds an 8-mer RNA containing a 2'-O-methyl group at its 3' end, using isothermal titration calorimetry (ITC). We found that the SUMO-tagged PAZ domain binds the ssRNA with a K_d of 4.0 μ M (new Fig. 5), indicating that the PAZ domain of Piwi

recognizes the piRNA 3' end, as in the other Argonaute proteins. Consistent with this, the residues that recognize the piRNA 3' end in the other PIWIs are highly conserved in Piwi (new Fig. S6). We have added the ITC data and the sequence alignment in the revised manuscript.

During the revision, we reported that the Piwi PAZ mutation (Y327A/Y328A) reduces the amounts of mature piRNAs in OSCs, highlighting the importance of the PAZ-mediated piRNA recognition for the piRNA maturation (Yamashiro *et al.*, *EMBO Rep*, 2019). Intriguingly, the length distribution of the Piwi-bound piRNAs (~23–30 nt with the peak of 26 nt) (Saito *et al.*, *Nature*, 2009) is wider than that of the Siwi-bound piRNAs (~27–29 nt with the peak of 28 nt) (Nishida *et al.*, *Cell Rep*, 2015) (Fig. L1). These observations suggest that the flexibility of the Piwi PAZ domain contributes to the accommodation of piRNAs with a wider range of lengths. We have added these statements with the references in the revised manuscript.

Saito et al. 2009. Nature
Figure S2b

Nishida et al. 2015. Cell Reports
Figure 1C

Figure L1. Length distributions of the piRNAs bound to Piwi and Siwi.

I am further not convinced that the structural data allows the authors to place the PAZ domain with confidence. The localization between two putative mercury ions can hardly be accurate. The authors report a decrease in Rwork from 0.253 to 0.244, a rather modest improvement and only meaningful if the Rfree does significantly improve as well. Figure 2C suggest for the reader that there is a drastically different conformation of the PAZ domain in Siwi and Piwi and this is

not supported sufficiently through the authors data. I suggest to not build the PAZ domain in the model and to indicate its position in Figs 1 and 2 with a schematic representation.

As the reviewer pointed out, it is difficult to model the PAZ domain into the relatively poor density without any prior information. Nonetheless, we would like to include the PAZ domain in the final model, due to the following reasons. The PAZ domain of Piwi shares significant sequence identity (39%) with that of Siwi, indicating that the PAZ domain of Piwi is structurally similar to that of Siwi. Indeed, the Phyre2 server generated a homology model of the Piwi PAZ domain with 100% confidence, based on the structure of the Siwi PAZ domain (PDB: 5GUH). Thus, we fitted this homology model into the electron density using rigid-body fitting, and refined the model using the secondary-structure restraints. The resulting PAZ domain model fit well to the omit electron density map, with the real space correlation coefficient of 0.6. In addition, the location of the PAZ domain is consistent with the anomalous difference peaks of the two mercury ions bound to the two cysteine residues in the PAZ domain. Furthermore, the R_{free} value decreased from 0.2628 to 0.2607, when the PAZ domain was included in the final model. Thus, we would like to include the PAZ domain in the final model, although the PAZ domain is flexible and not perfectly resolved in the electron density map. We have added statements about the flexible conformation of the PAZ domain in the revised manuscript, so the readers understand that the PAZ domain of Piwi does not adopt a fixed conformation that is drastically different from that of Siwi.

2. Phenotype of 'slicer-Piwi'

The authors need to comment on their earlier findings (2006 Genes Dev) that recombinant Drosophila Piwi is an active slicer. Despite this earlier report, other authors have suggested previously that Piwi would not have any RNA cleavage activity (based on the imperfect catalytic tetrad).

We appreciate the important suggestion. In this study, we purified FLAG-tagged Piwi from OSCs using anti-FLAG antibody beads, and measured its cleavage activity toward the ssRNA substrate. We found that Piwi does not cleave the ssRNA substrate under our experimental conditions. Importantly, we also found, in contrast to wild-type Piwi, the slicer-Piwi mutant cleaves the ssRNA substrate with efficiency comparable to that of Siwi. Based on these results, we concluded that Piwi is not a slicer. In our previous study, we purified GST-tagged Piwi from *E. coli* using Glutathione Sepharose resin, and used an excess amount (~1 μg) of the partially

purified Piwi for *in vitro* cleavage experiments (Saito *et al. Genes Dev*, 2006). Thus, it is likely that we detected the negligible, if any, slicer activity by Piwi in the previous study. We added these statements in the revised manuscript.

The crystal structure together with biochemical analysis presented here proves that nicely. The fact that the authors are able to generate 'slicer-Piwi' through the mutation of key residues demonstrates that the inactivity of Piwi indeed is based on the altered catalytic tetrad and its environment. This is an interesting experiment. In their rescue assay in OSC cells, the authors demonstrate that 'slicer-Piwi' does complement a Piwi knockdown and that 'slicer-Piwi' is able to initiate co-transcriptional transposon silencing (this is shown only indirectly as silencing is still Gtsf1 dependent). Based on in vitro experiments using target RNA that has impaired complementarity, the authors state that "slicing activity could compromise the Piwi-mediated co-transcriptional silencing" (lines 97-98). To strengthen their case, the authors should consider analyzing the effect of 'slicer-Piwi' in a fly model, rather than relying on the fairly rough rescue assay in cell culture. Only a fly rescue experiment will allow a strong statement about whether or not slicing does affect Piwi function.

Thank you for the critical comment. We agree with the reviewer that an analysis of slicer-Piwi in a fly model will be required to fully clarify the biological significance of the slicer activity. Nonetheless, as the editor suggested, an analysis in a fly model seems beyond the scope of this work, and we hope that future work using a fly model will clarify the effects of the slicer activity on the Piwi-mediated co-transcriptional silencing.

Reviewer #2 (Remarks to the Author):

Piwi protein, identified in 1997 as a critical gene for germline stem cell division, binds piRNA to form piRNA-induced silencing complexes that silence transposons and maintain genome integrity. PIWI and AGO are two clades of Argonaute proteins. The structures of AGO proteins have been well studied in the last 15 years. However, the structural information of Piwi protein is very limited due to difficulties in sample preparation. To date, only one Piwi protein Siwi structure has been reported.

Yamaguchi et al. solved the crystal structure of Drosophila Piwi protein in complex with endogenous piRNAs. This is the second structure of the Piwi protein. Although the overall structure of Piwi is similar to Siwi protein, the structure of the Piwi-piRNA complex reveals

some PIWI specific structural features including the overall domain arrangement and metal-dependent piRNA binding. Authors find that the Piwi protein has a non-canonical DVDK tetrad and lacks the slicer activity, but the DEDH mutant has the slicer activity. The authors propose that Piwi protein lack of slicer activity serves as a piRNA-guided RNA binding platform, ensuring faithful co-transcriptional silencing of transposons. Collectively, this study provides significant insights into the molecular mechanism of the Piwi-mediated transposon silencing.

We appreciate the positive comments by the reviewer.

Piwi proteins and prokaryotic Ago proteins share several structural features. 1. The binding of the 5'-phosphate of piRNA and siRNA are both metal-dependent. 2. The Glu finger from both the Piwi and prokaryotic Ago proteins are located away from the other three catalytic residues. Authors compared the structures of Piwi, Siwi, and hAgo2, but comparisons between Piwi and PfAgo and TtAgo are limited. Additional comparisons between the Piwi and Pf/TtAgo are suggested.

According to the reviewer's comment, we have added a structural comparison of Piwi with the prokaryotic AGOs, such as PfAgo and TtAgo, in terms of their piRNA recognition and catalytic tetrad in the revised manuscript (new Fig. S7).

In page 8 lines 215 and 216, Fig. 5A should be Fig. 5B, and Fig. 5B should be Fig. 5A.

Thank you for the comment. We have corrected it.

Figures 2C, 2D and S4 should be optimized to improve their clarity.

According to the reviewer's comment, we have modified Figs. 2C, 2D, and S4 to improve their clarity.